# Baited reconstruction with 2D template matching for high-resolution structure determination in vitro and in vivo without template bias

Bronwyn A Lucas[1,2,3]*, Benjamin A Himes[1,4], Nikolaus Grigorieff[1,4]*

[1]RNA Therapeutics Institute, University of Massachusetts Chan Medical School, Worcester, United States; [2]Department of Molecular and Cell Biology, University of California Berkeley, Berkeley, United States; [3]Center for Computational Biology, University of California Berkeley, Berkeley, United States; [4]Howard Hughes Medical Institute, Chevy Chase, United States

**Abstract** Previously we showed that 2D template matching (2DTM) can be used to localize macromolecular complexes in images recorded by cryogenic electron microscopy (cryo-EM) with high precision, even in the presence of noise and cellular background (Lucas et al., 2021; Lucas et al., 2022). Here, we show that once localized, these particles may be averaged together to generate high-resolution 3D reconstructions. However, regions included in the template may suffer from template bias, leading to inflated resolution estimates and making the interpretation of high-resolution features unreliable. We evaluate conditions that minimize template bias while retaining the benefits of high-precision localization, and we show that molecular features *not* present in the template can be reconstructed at high resolution from targets found by 2DTM, extending prior work at low-resolution. Moreover, we present a quantitative metric for template bias to aid the interpretation of 3D reconstructions calculated with particles localized using high-resolution templates and fine angular sampling.

*For correspondence:
bronwynlucas@berkeley.edu (BAL);
niko@grigorieff.org (NG)

## eLife assessment

This is an **important** demonstration of how the false-positive rate of high-resolution 2D template matching to find particles of a given target structure in 2D cryo-EM images (2DTM) relates to overfitting the data towards the template. The authors present new methods to measure the amount of model bias that gets introduced in high-resolution features of such maps, with **compelling** evidence that high-resolution features that are not present in the template can still be reconstructed in 3D from images obtained by 2DTM.

## Introduction

Over the last decade, single-particle cryogenic electron microscopy (cryo-EM) has emerged as a high-resolution technique to study molecules and their assemblies in a near-native state (*Guaita et al., 2022*). In the most favorable cases, close to 1 Å resolution can be achieved, rivaling results obtained by protein crystallography (*Nakane et al., 2020*; *Yip et al., 2020*). The resolution obtained from a single-particle dataset depends on the quality of the images, the accuracy of particle alignment and imaging parameters, the structural integrity of the sample, and the number of particles contributing to a reconstruction. For a high-quality dataset, between 20,000 and 70,000 asymmetric units of

well-aligned and homogeneous particles have to be averaged to reach sub 2 Å resolution (*Nakane et al., 2020*; *Yip et al., 2020*). Methods development in a related cryo-EM technique has also enabled imaging particles in situ at high resolution using tomography and subtomogram averaging. These in situ subtomogram averages now approach 3 Å resolution (*Tegunov et al., 2021*), a resolution obtained routinely for single-particle reconstructions in vitro. Data collection for tomography requires more time compared to the single-particle technique due to the need for acquiring a tilt series, and processing tends to be computationally more expensive due to the additional degrees of freedom, compared to 2D images used in the single-particle technique. An additional complication of averaging images of molecules for in situ structure determination is the selection of valid targets, which have to be identified against a background of other molecules inside the cell or tissue being imaged. This is in contrast to a typical single-particle dataset, in which the particles have undergone a purification step that enriches the particle of interest, and imaged in solvent which makes particle selection more reliable.

2D template matching (2DTM) is an approach that can be used to identify target molecules and complexes in cryo-EM images of cells and cell sections, using single images of nominally untilted specimens (*Lucas et al., 2022*; *Lucas et al., 2021*; *Rickgauer et al., 2020*; *Rickgauer et al., 2017*). This approach can be used in combination with 3D template matching (3DTM) to identify targets in tomograms collected from the same areas imaged for 2DTM. 2DTM is fundamentally limited by the background generated by overlapping molecules in untiled views of the sample, imposing a size limit on what can be detected (*Rickgauer et al., 2017*). A combined approach of 2D and 3DTM could benefit from the strengths of both approaches, with better target detection (lower false negative rate) of 3DTM in the tomograms, and the improved overall precision of 2DTM in the untiled views (*Lucas et al., 2021*). In our previous studies, we have demonstrated that the targets detected using 2DTM can be used to calculate 3D reconstructions showing novel details not present in the template (*Lucas et al., 2022*; *Lucas et al., 2021*; *Rickgauer et al., 2017*). 3D reconstruction is straightforward because for every detected target, 2DTM also determines their $x,y$ location in the image, three Euler angles, and image defocus, that is, all the parameters needed to calculate a single-particle reconstruction. Using this approach, we revealed non-modeled density for the viral polymerase (VP1) bound to a rotavirus capsid (*Rickgauer et al., 2017*), for the small ribosomal subunit (SSU) and tRNAs (*Lucas et al., 2022*; *Lucas et al., 2021*), as well as structural differences between *Mycoplasma pneumoniae* and *Bacillus subtilis* large ribosomal subunits (LSUs) (*Lucas et al., 2021*). Interpretation of reconstructions obtained from 2DTM targets can be hindered by template bias (*Lucas et al., 2022*; *Lucas et al., 2021*), that is, the reproduction of modeled features included in the template, that are reproduced in the reconstructions, even though they do not correspond to structural features in the detected particles. This could result from inclusion of pure noise particles and/or local overfitting of particle parameters in the presence of signal. Our previous studies showed that template bias does not prevent the discovery of new structural features at low resolution that were not represented by the template, but it has yet to be determined if this is true for high-resolution features, which are more susceptible to noise overfitting (*Stewart and Grigorieff, 2004*). 2DTM could in principle be used to study the structure of targets at high resolution, that would otherwise be too small to identify on their own, as long as they bind or are otherwise rigidly attached to a larger target that can be located by 2DTM. However, due to limitations in the number and heterogeneity of particles in previous studies, it was unclear whether this approach could indeed recover reliable high-resolution information.

In the present study, we explore this possibility further to assess the resolution that can be obtained in unmodeled regions omitted from the template. We analyze a published single particle dataset of β-galactosidase (Bgal) using 2DTM, and 60S LSUs detected in images of *Sacchromyces cerevisiae* lamellae. In both cases, we show high resolution in areas of the reconstruction that were omitted in the template, demonstrating the utility of 2DTM for structure discovery. We present a new metric to quantify template bias in a template-based 3D reconstruction, making reconstruction from 2DTM targets a more broadly useful tool.

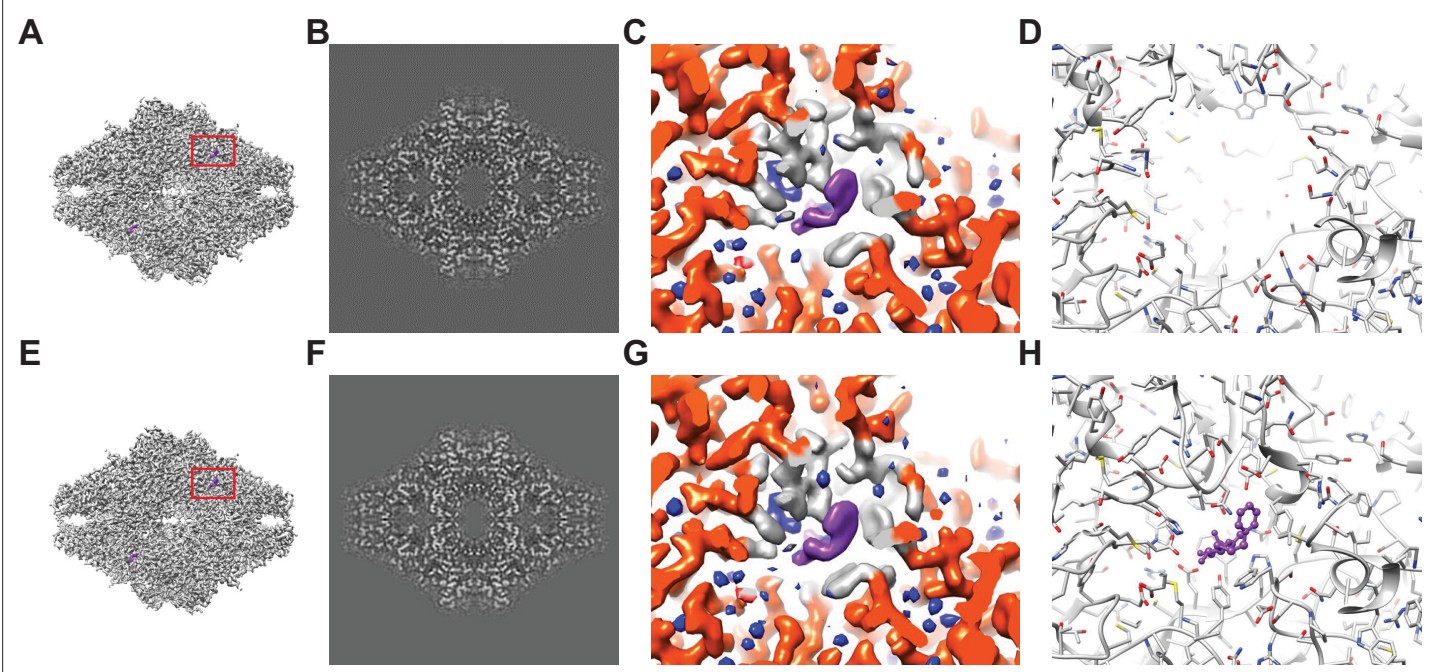

**Figure 1.** Baited reconstruction for visualization of β-galactosidase (Bgal) ligand binding pocket at high resolution. (**A**) Reconstruction of Bgal from 2DTM coordinates using images from a previously published dataset (*Saur et al., 2020*) using a Bgal crystal structure (PDB: 1DP0) (*Juers et al., 2000*) as a template, with a 10 Å sphere around the phenylethyl β-D-thiogalactopyranoside (PETG) ligand omitted. (**B**) A 2D slice through the reconstruction in (**A**) including the region deleted from the density shows no obvious discontinuity in the density. (**C**) A view of the density in (**A**) indicated with a red box, with regions within 1.8 Å of the template model highlighted in red. Gray indicates density of Bgal outside of the template, purple indicates density consistent with the position of PETG, and blue indicates additional density that likely represent water molecules. (**D**) A stick diagram showing the locations of the atoms in the template used for template matching. (**E**) Published density from *Saur et al., 2020* aligned and scaled as in (**A**). (**F**) As in (**B**), showing a region of the published density in (**E**). (**G**) As in (**C**), showing the same region of the published density in (**E**). (**H**) As in (**D**), showing all atoms annotated in the crystal structure, including those omitted before generating the 2DTM template.

The online version of this article includes the following figure supplement(s) for figure 1:

**Figure supplement 1.** Results from a 2D template matching search of an image of Bgal with bound PETG using an apo structure of Bgal as a template.

## Results

### Reconstruction of the Bgal ligand binding pocket

To show the potential of 2DTM to reveal new structural details at high resolution, we analyzed a published single-particle cryo-EM dataset of *Escherichia coli* Bgal bound to phenylethyl β-D-thiogalactopyranoside (PETG) (*Saur et al., 2020*). The dataset was used previously to calculate a 2.2 Å single-particle reconstruction (EMDB-10574) that displays density for a number of specifically bound water molecules in the structure, including in the PETG binding pocket. The authors also built an atomic model into the high-resolution map (PDB: 6TTE). For our template, however, we used an atomic model of ligand-free Bgal determined by X-ray crystallography at 1.7 Å (PDB: 1DP0) (*Juers et al., 2000*). Using a model that was built into a map that is independent from the data analyzed by 2DTM aids our demonstration of 2DTM as a tool that can make use of atomic models experimentally unrelated to the data being analyzed.

To demonstrate high resolution in areas omitted in the template, we removed atoms in the vicinity of all D2 symmetry-related ligand binding pockets, within a 10 Å radius centered on the side chain amide nitrogen atom of asparagine 102 (in PDB: 1DP0). The truncated atomic model was used to generate a template with *cis*TEM's simulator (*Himes and Grigorieff, 2021*) (see Materials and methods). We searched 558 micrographs downloaded from the EMPIAR database (EMPIAR-10644) and obtained 59,259 targets with 2DTM SNRs above a threshold of 7.3 (*Figure 1A*), the standard threshold calculated to limit the average number of false positives (false positive rate) to one per micrograph, based on the given search parameters and a Gaussian noise model (*Rickgauer et al., 2017*) (see *Equation 2* below). To reduce the particles to a number closer to the final dataset used

to calculate the 2.2 Å cryo-EM map in (49,895), and to enrich for the particles most similar to the template (similar to selecting the best classes in *Saur et al., 2020*), we limited our targets to those with 2DTM SNRs above 9.0 and obtained a final dataset of 55,627 particles.

The identified 55,627 targets were extracted together with their template-matched $x,y$ positions, Euler angles, and CTFFIND4-derived defocus values using prepare_stack_matchtemplate (*Lucas et al., 2021*), and the particle stack and alignment parameters were imported into *cis*TEM as a refinement package for further single particle processing. The Fourier shell correlation (FSC) (*Harauz and van Heel, 1986*) for the initial reconstruction calculated from the template-matched alignment parameters indicated a resolution of 2.4 Å (FSC = 0.143) (*Rosenthal and Henderson, 2003*). We performed further refinement against the template while keeping the refinement resolution limit of 3.0 Å – one cycle of defocus and beam tilt refinement, followed by a refinement of alignment parameters and another cycle of defocus and beam tilt parameters. The final reconstruction (*Figure 1A–C*) displayed a resolution according to the FSC of 2.2 Å (*Figure 1—figure supplement 1F*). As mentioned above and previously discussed (*Lucas et al., 2021*), resolution estimates based on the FSC can be affected by template bias. Therefore, the present estimate has to be considered unreliable, and has to be supported by additional evidence, such as inspection of features visible in the density map.

Peaks corresponding to detected targets are clearly visible (*Figure 1—figure supplement 1B*). The average 2DTM SNR for this dataset was 11.6, and a maximum of 16.3, which is in the range of what is expected for a 465 kDa target (*Rickgauer et al., 2020*; *Rickgauer et al., 2017*). The refined reconstruction shows clear density for PETG and water molecules in the ligand binding pocket that were omitted in the template (*Figure 1C*). Comparison of this reconstruction with the published map (*Figure 1G*) suggests that they are virtually identical and that there is little or no evidence of template bias in the 2DTM reconstruction. An assessment of the local resolution using Phenix (*Liebschner et al., 2019*) further indicates a resolution of about 1.8 Å in the binding pocket, consistent with the clear density for water.

## Baited reconstruction visualizes ribosomes at near atomic resolution in FIB-milled lamellae

To investigate whether 2DTM can be used to generate reliable high-resolution reconstructions from images derived from cellular samples, we used a previously published dataset of 37 images of four FIB-milled lamellae generated from *S. cerevisiae* cells treated with the translation inhibitor cycloheximide (CHX) to enrich the ribosome population in a single state (*Figure 2A*; *Lucas and Grigorieff, 2023*). The lamella samples were not tilted during data collection and therefore exhibited a small tilt with respect to the electron beam of about 8° due to the milling angle during sample preparation. We identified 12,210 LSUs with 2DTM in the cytoplasm using a threshold of 7.85 which corresponds to an expectation of one false positive per image across the dataset, or ~0.3% of the particles (*Figure 2—figure supplement 1A–C*). Local positional and orientational refinement was performed using the *cis*TEM program refine_template (*Lucas et al., 2021*) and the original template as a reference. The refined 2DTM coordinates were used to calculate an initial reconstruction with a nominal resolution of 3.15 Å (FSC = 0.143) (*Rosenthal and Henderson, 2003*). One cycle of beam tilt refinement against the reconstruction improved the resolution to 3.1 Å (*Figure 2A and B*). Unlike for the in vitro Bgal reconstruction, further refinement of the other alignment parameters using the reconstruction as a reference caused the resolution to decrease to 8 Å. The reconstruction has reduced signal at high spatial frequencies relative to the template as indicated by the half-map FSC (*Figure 2—figure supplement 1D*). This, combined with the higher background and lack of low-resolution contrast of the ribosomes in the cellular lamella relative to a purified sample, may reduce the alignment accuracy relative to the high-resolution 2DTM template. This highlights the importance of high spatial frequencies for alignment of particles in images with strong background, in contrast to images of purified samples that show strong low-resolution features, which are important for reliable particle alignment (*Stewart and Grigorieff, 2004*).

As previously reported (*Lucas et al., 2022*; *Lucas et al., 2021*), we found density consistent with the SSU and tRNAs, that did not derive from the template. In the present case, local resolution estimation shows that parts of the SSU are resolved at <4 Å resolution (*Figure 2B*). The SSU is conformationally variable and shows considerable positional heterogeneity relative to the LSU. Therefore, this value is likely an underestimate of the potential attainable resolution in reconstructions from 2DTM targets

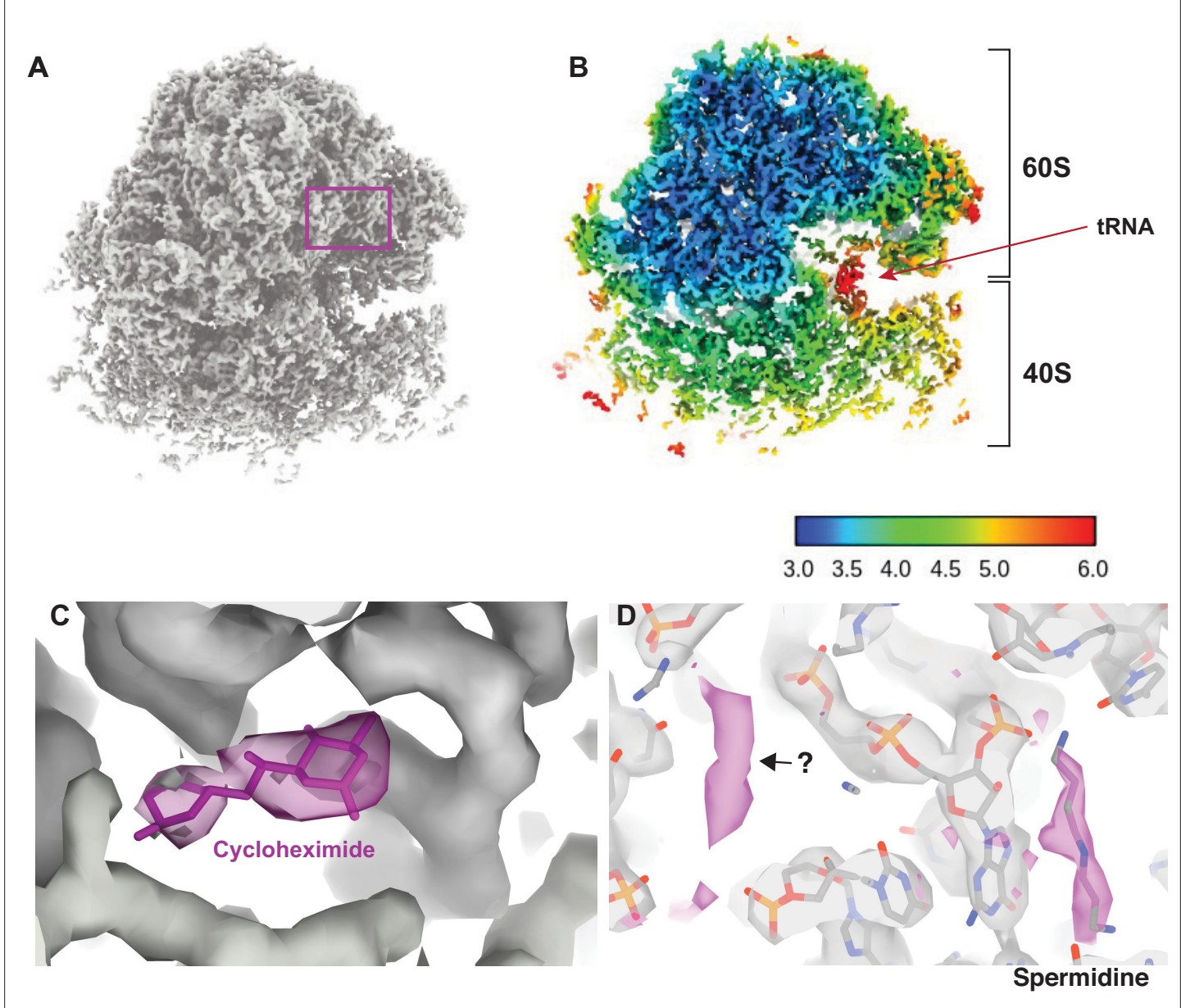

**Figure 2.** Visualizing drugs and small molecules bound to the ribosome in vivo. (**A**) A reconstruction of the ribosome from 2DTM coordinates identified in the cytoplasm of FIB-milled *S. cerevisiae* cell sections showing clear density for both the 60S (part of the template) and the 40S (outside of the template). (**B**) A slice of the reconstruction in (**A**), indicating the local resolution using the indicated color coding. The arrow indicates the P-site tRNA. (**C**) Regions of the density >3 Å from the template model are indicated in pink. The crystal structure PDB: 4U3U was aligned with the template model and the position of cycloheximide (CHX) was not altered. (**D**) As in C, showing density corresponding to a spermidine (PDB: 7R81) and unaccounted for density outside of the template (black arrow), which may also represent a polyamide.

The online version of this article includes the following figure supplement(s) for figure 2:

**Figure supplement 1.** Results from a 2D template matching search of a yeast lamella using the yeast 60S ribosomal subunit as a template.

**Figure supplement 2.** Cycloheximide stalls the ribosome in a non-rotated PRE translocation state in vivo.

in cells. Although the map represents an average of all states identified, we observed clear density for tRNAs in the A/A and P/P state with apparent density for the polypeptide on the A site tRNA (*Figure 2—figure supplement 2*) and no clear density for E-site tRNAs. This allowed us to conclude that CHX stalls ribosomes in the classical PRE translocation state in vivo, likely by preventing transition of the P/P tRNA to the P/E state consistent with an in vitro structure of the translating *Neurospora*

*crassa* ribosome (*Shen et al., 2021*) and inference from ribosome profiling data (*Lareau et al., 2014*; *Wu et al., 2019*). The relatively low resolution of the tRNAs likely reflects the mixed pool of tRNA depending on the codons on which the ribosome stalled as well as a mixture of states.

### Visualization of drug-target interactions in cells

Drug-target interactions can be visualized at high-resolution in vitro with cryo-EM and X-ray crystallography. However, it is unclear whether this recapitulates the binding site in vivo, possibly missing weak interactions that are disrupted during purification. Visualizing drug-target interactions in cells is therefore an important goal. We observed additional density near the ribosomal E site not present in the template that is consistent with the position of CHX in a previously published crystal structure (*Garreau de Loubresse et al., 2014*; *Figure 2C*). The density was sufficiently well resolved to dock CHX and provide in vivo confirmation for the position and orientation of CHX binding in the E site.

We noted several key differences between the model built from the in vitro CHX-bound structure and the in situ CHX-bound structure. First, we did not observe density for eIF5A but did observe density consistent with binding of spermidine (*Figure 2D*), as has been observed previously for the in vitro CHX-bound *N. crassa* ribosome (PDB: 7R81) (*Shen et al., 2021*). This demonstrates that spermidine can bind to ribosomes within cells, however, whether spermidine binds as part of the translation cycle or whether stalling of translation with CHX allowed for spermidine to bind is unclear. Baited reconstruction with 2DTM could be used to further probe the function of polyamides to regulate translation in vivo.

Baited reconstruction using the LSU as a template model allowed us to visualize the binding of small molecules such as drugs and polyamides to the ribosome within cells. This demonstrates the power of baited reconstruction to reveal biologically relevant features that would only be evident at high resolution.

### Omit templates reveal high-resolution features without template bias

The local resolution of parts of the LSU were measured at ~3 Å, however, this region overlapped with the template and therefore the resolution measure using standard tools may be unreliable. To assess the resolution in this region we repeated this experiment with a template that lacked the ribosomal protein L7A. Since this protein was not present in the template, any density in this region cannot be due to template bias. We found that the local resolution of L7A was indistinguishable from the surrounding density and showed varying local resolution from 3.2 to 4.5 Å (*Figure 3A–B*). The density was sufficiently well resolved to observe side chains in regions that were lacking from the template (*Figure 3C*). This suggests that baited reconstruction with 2DTM coordinates can be used to generate high-resolution reconstructions from cellular samples, free from template bias, and demonstrates an approach to verify local resolution estimates.

To examine the recovery of high-resolution information with single residue precision, we generated another truncated template by removing every 20th residue from each chain. This resulted in a total reduction of 51 kDa or ~3% of the template mass. We then localized 12,090 targets using the same 2DTM protocol as for the full-length template. The small difference in template mass minimally affected target detection, only 120 targets (<1%) were missed, and there were minimal deviations in the locations and orientations for the remaining targets. The 3D reconstruction generated from the detected targets showed clear density corresponding to nucleotides (*Figure 3D and E*) and various amino acids (*Figure 3F–K*) that were missing from the template and therefore cannot derive from template bias. This demonstrates that omitting small randomly scattered regions from a 2DTM template can be used to assess template bias throughout the reconstruction.

### Quantifying template bias

The calculation of reconstructions from targets identified by 2DTM, which relies on a priori structural models, bears the danger of generating results that reproduce features of the template even when these features are absent from the targets to be detected. In the field of cryo-EM, this is often referred to as the 'Einstein from noise' problem (*Henderson, 2013*). The risk of template bias increases with dataset size (number of images), as well as the ratio of false positives vs true positives. Template bias in reconstructions generated from 2DTM targets is generally avoided because the scoring function (SNR threshold) is set to reject most false positives. To quantify template bias in reconstructions at

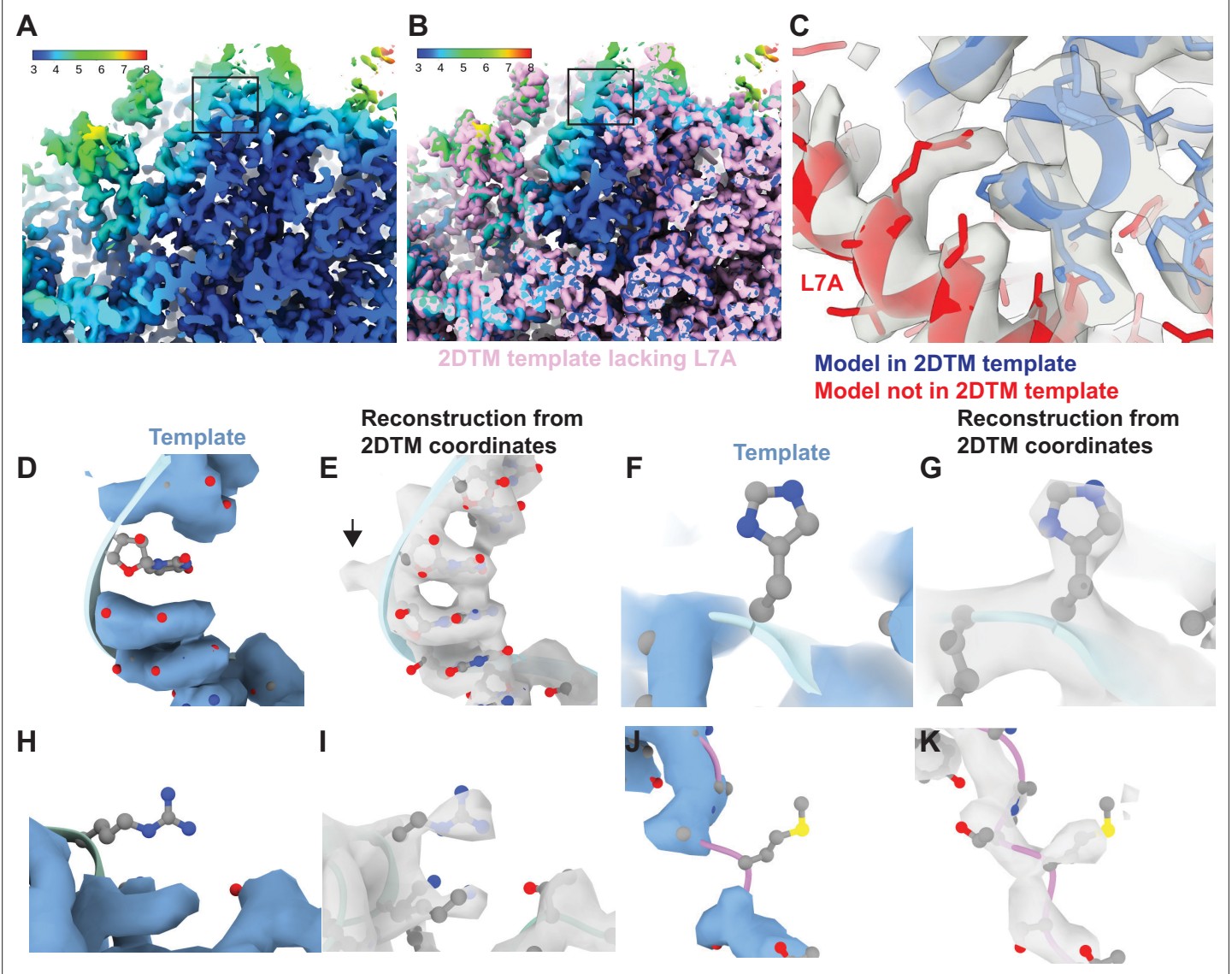

**Figure 3.** Baited reconstruction reveals high-resolution features in vivo without template bias. (**A**) Slice of a reconstruction using 2DTM coordinates identified with a template lacking the protein L7A. Color coding indicates the local resolution as indicated in the key. (**B**) As in (**A**), pink indicates the 2DTM template used to identify the targets used in the reconstruction. (**C**) The model PDB: 6Q8Y is shown in the density. Red corresponds to the protein L7A, which was omitted from the template used to identify targets for the reconstruction. Blue corresponds to model features that were present in the template. (**D**) Single nucleotide omit template and (**E**) reconstruction showing emergence of density outside of the template, including a phosphate bulge, black arrow. Single amino acid omit templates lacking Phe (**F**), Arg (**H**), or Ser (**J**) and density (**G, I, K**), respectively, showing emergence of features consistent with each amino acid.

various 2DTM SNR thresholds, we generated a series of reconstructions at different thresholds using targets identified with a full-length LSU template ('full' template) and the template lacking 3% of the residues ('omit' template) covering different areas of the model, while retaining most detections relative to the full template as described above (**Figure 4A**). We wrote a new *cis*TEM program measure_template_bias (see Materials and methods) that calculates the difference between map densities $\rho_{full}$ and $\rho_{omit}$ in these reconstructions, in the omitted regions:

$$\Omega = \frac{\rho_{full} - \rho_{omit}}{\rho_{full}} \tag{1}$$

As expected, for high 2DTM SNR thresholds (few or no false positives), the template bias $\Omega$ was only a few percent, while for lower thresholds, it approached 100% (**Figure 4A**). This was consistent

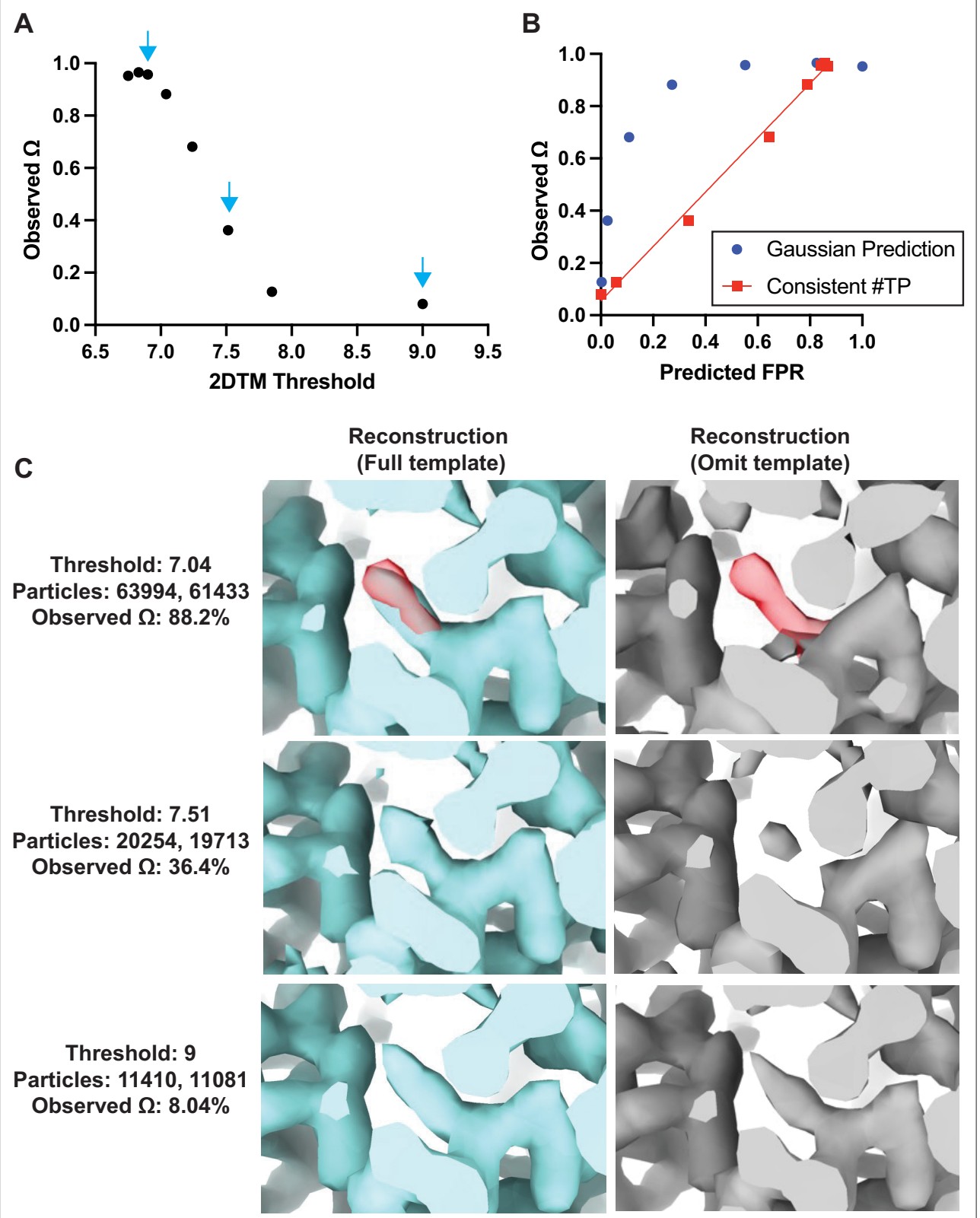

**Figure 4.** Baited reconstruction provides a quantitative metric for template bias. (**A**) Observed template bias ($\Omega$) calculated using the *cis*TEM program measure_template_bias as a function of the 2DTM SNR threshold used to select targets from images of yeast lamellae. Blue arrows indicate the reconstructions shown in **C**. (**B**) Plot showing a comparison of the predicted false positive rate and the observed $\Omega$. The plotted straight line indicates

*Figure 4 continued on next page*

*Figure 4 continued*

the best fit linear function $y = 0.96x - 0.05$. (**C**) Images showing the same region of maps resulting from reconstruction using targets identified with the indicated template at the indicated 2DTM SNR threshold. Red indicates the location of the omitted residue in the omit template.

The online version of this article includes the following figure supplement(s) for figure 4:

**Figure supplement 1.** Plot showing the average relative number of large ribosomal subunits (LSUs) detected in seven images of ~150 nm thick lamellae using templates generated from subsections of the LSU of the indicated molecular mass.

with increased density in the reconstructions using the full template relative to the omit template (*Figure 4C*). The observed lower limit of Ω of ~8% (*Figure 4A*) is likely due to some overfitting of noise when template-matching true particles, rather than inclusion of false positives. This overfitting may manifest itself in small alignment errors of the targets against the matching template, and a bias of these errors toward compensating for any mismatch between target and template, such as omitted regions in the template. Further work to quantify Ω at different spatial frequencies will be informative to assess the contribution of local overfitting to template bias.

If we assume that the template bias is proportional to the rate of false positive detection, $r_f$ , we can plot the expected false positive rate, $r_{f,model}$ , against the observed template bias Ω (*Figure 4B*). The expected false positive rate is given by the complementary error function (*Rickgauer et al., 2017*) as

$$r_{f,model} = \frac{1}{2} erfc \left( \frac{SNR_t}{\sqrt{2}} \right) \tag{2}$$

where $SNR_t$ is the 2DTM SNR threshold applied to the template search results. The plot shows that the template bias is not proportional to the expected false positive rate (*Figure 4B*). This is likely due to the variable background found in images of lamellae, which means that the spectral whitening that is applied to the images before the search (*Rickgauer et al., 2017*) does not whiten all areas of the images evenly. This results in local deviations of the background (noise) distribution from the Gaussian noise model implied in *Equation 2*, leading to higher-than-expected false positive ratios at low SNR thresholds.

If we estimate the number of true targets at 13,456 (the number of targets identified by both templates at a threshold of 7.85) and recalculate the number of false positives as the overall number of detected targets in excess of this number, the template bias is approximately proportional to the false positive rate (red line in *Figure 4B*). Further work is required to develop an improved noise model that predicts the correct number of false positives in images of variable contrast, such as images of cellular lamellae. Furthermore, it is important to note that the 2DTM SNR threshold used here to exclude most of the false positives also leads to a rejection of true positives. The number of these false negatives depends on the 2DTM SNR generated by the targets, which is proportional to their molecular mass (*Figure 4—figure supplement 1*). For our data and 150 nm thick lamellae, this means that targets below about 300 kDa will not be detected. Improvements in cryo-EM instrumentation, sample preparation, image processing, and 2DTM methods will lower this limit (*Russo et al., 2022*).

## Discussion

We show here that baited reconstruction with 2DTM can reveal high-resolution detail in regions not modeled in the template. Using a previously published single-particle dataset we observe interactions between specific side chains with water and a ligand. Using particles localized in FIB-milled yeast lamellae, we observe specific binding of the drug CHX and polyamides to the ribosome in cells. We show that baited reconstruction can be used to recover high-resolution features in cells without template bias in regions omitted from the template, and quantify template bias in regions overlapping the template. The use of 2D images to generate high-resolution reconstructions makes this process significantly faster and less computationally expensive relative to tomography. Baited reconstruction is analogous to a 'pulldown' assay in molecular biology, wherein a 'bait' molecule is used to capture and identify novel interacting 'prey'. This strategy is distinct from prior structure determination strategies because it makes use of a high-resolution template, traditionally avoided to prevent introducing template bias artifacts (*Henderson, 2013*). Baited reconstruction leverages the advantages of precise

targeting with a high-resolution template, while avoiding the template bias by focusing on regions omitted from, or external to the template. Baited reconstruction can therefore leverage the wealth of existing structural data, as well as molecular models generated by the newly available structure prediction tools (*Baek et al., 2021*; *Evans et al., 2022*; *Jumper et al., 2021*), to approach biological and pharmacological questions in vitro and in vivo.

## Implications for drug discovery

One of the most direct applications of this approach is to drug discovery. During the drug development pipeline, potentially thousands of variants of a lead compound are tested relative to a single protein target. Determining the structures of each in complex with its protein partner using the traditional single-particle cryo-EM workflow can be time-consuming and laborious, and often requires image processing expertise. The strategy presented here could be used to streamline this process substantially.

The ribosome is a major target of antibiotic and anticancer drugs. We have demonstrated that baited reconstruction with 2DTM can reveal drug-ribosome interactions directly in cells. The reconstructions are at comparable resolution relative to the state-of-the-art from tomography, while using a more streamlined data collection and processing pipeline that could be easily automated. This approach could therefore be used to more efficiently characterize the mechanism of action of antibiotic drugs directly in cells. Since 2DTM does not require purification, the interactions with other cellular complexes can also be investigated.

## 2DTM accelerates high-resolution in situ structure determination

Baited reconstruction is substantially faster and a more streamlined pipeline for in situ structure determination compared to cryo-ET and subtomogram averaging. Current pipelines for in situ structure determination using cryo-ET and subtomogram averaging are time-consuming and require expert knowledge to curate an effective pipeline. We expect focused classification to identify sub-populations to further improve the resolution of in situ reconstructions from 2DTM targets. To help classify particles against a cellular background without introducing alignment errors (see Results), alignment parameters (Euler angles, $x,y$ shifts) can remain fixed. While tomograms are required to provide the cellular 3D context of molecules, our work shows that it is not always necessary to use tomography to generate high-resolution reconstructions of macromolecular complexes in cells. 2DTM could reduce the manual effort and time for structure determination in cells when compared to subtomogram averaging, depending on the time it takes to annotate, refine, and classify the subtomograms.

Our approach also differs from the recently published isSPA method (*Cheng et al., 2023*; *Cheng et al., 2021*). isSPA follows the traditional single-particle workflow, applied to particles in their native environment. Particles are selected using a template that is limited to an intermediate resolution of 8 Å, resulting in an initial particle stack that contains many false positives. Selection of the top scores, followed by standard single-particle classification and alignment protocols, then yields reconstructions of the detected targets. This approach is particularly successful in situations where there is a high concentration of the particle of interest, such as Rubisco inside the carboxysome (*Cheng et al., 2021*), capsid proteins in viral capsids (*Cheng et al., 2021*), and phycobilisome and photosystem II in the thylakoid membranes inside *P. purpureum* cells (*Cheng et al., 2023*). In contrast, using 2DTM, we select targets with high fidelity, avoid false positives, and determine the molecule pose to high accuracy without the need for an intermediate reconstruction from the detected targets to act as reference for further refinement. By using the full resolution of the signal present in the images, 2DTM is also more sensitive than isSPA, detecting particles of 300 kDa in 150 nm lamellae (*Figure 4—figure supplement 1*). These differences mean that 2DTM can be used with fewer, and potentially smaller particles to achieve high-resolution structures compared to isSPA and other techniques following the canonical single-particle averaging workflow. As demonstrated here, the detection criterion used in 2DTM largely avoids overfitting artifacts in reconstructions by eliminating images that are not statistically distinguishable from noise. This makes 2DTM particularly useful for in situ structure determination, which is often limited by the low abundance of the target complexes inside the cell. By reducing the number of particles needed to achieve high-resolution reconstructions in cells, baited reconstruction with 2DTM will make it possible to determine the structures of less abundant complexes in cells.

## Application to in vitro single-particle analysis

Our results of a single-particle dataset of purified Bgal demonstrates another use case for 2DTM. In the original analysis of this dataset using the traditional single-particle workflow, 136,013 particles were initially selected using template-based particle picking (Gautomatch 0.56, http://www.mrc-lmb.cam.ac.uk/kzhang/) (*Saur et al., 2020*). 2D classification, ab initio reconstruction, and further 3D classification eventually yielded a 2.2 Å reconstruction showing the bound ligand (PETG). The same result was achieved with a simple run of 2DTM, without requiring manual intervention or expert knowledge in the image processing workflow. In a separate 2DTM search using the first 277 images of the dataset and a crystal structure of GroEL (PDB: 1GRL) as a template – a particle of comparable size to Bgal – we detected only 53 targets above the default SNR threshold (excluding two images that had sharp black lines across them), and none above a threshold of 9.0. This further demonstrates the high level of discrimination of 2DTM between true and false positives, as shown earlier (*Rickgauer et al., 2017*). Besides the streamlined workflow, 2DTM can therefore also be used in the presence of impurities to reliably select the particles of interest. Using multiple templates, particles could be classified to arrive at quantitative estimates of particles occupying defined conformational states. The reduced need for sample purity and dataset size to perform such analyses may further accelerate the 2DTM workflow, compared to the traditional single-particle workflow, provided appropriate templates are available.

Furthermore, validation of map and model quality is a major challenge in cryo-EM. Current methods use low-pass filtered templates to avoid template bias at high spatial frequencies. We here present a quantitative estimate of local and global template bias in sequence space. This will allow the full resolution of the template to be used to localize particles more specifically and avoid false positives. This may assist in identification of particle classes in the localization stage and can streamline the reconstruction process. Estimating template bias with baited reconstruction can provide a quantitative metric of map and model quality that may find broad utility in single particle and in situ workflows.

## Application to subtomogram averaging

Recently, higher resolution template matching and finer angular sampling have also been explored for the analysis of cryo-ET 3D reconstructions (*Chaillet et al., 2023*; *Cruz-León et al., 2023*). This approach has clear advantages because it reduced false positives due to low-resolution overlap (*Chaillet et al., 2023*; *Cruz-León et al., 2023*) and provides more specific localization of targets in a crowded cellular environment. However, if the identified targets are subsequently used for subtomogram averaging, the reconstructions may exhibit template bias. Both baited reconstruction and the quality metrics we describe above could be applied to subtomogram averaging pipelines.

## Future applications

We have shown that it is possible to recover single residue detail, and even the location of water molecules in the most favorable cases, using baited reconstruction with cryo-EM. This approach is analogous to the use of OMIT maps in X-ray crystallography to avoid model bias (*Bhat and Cohen, 1984*; *Hodel et al., 1992*) and the M-free score used to estimate reference bias in subtomogram averaging (*Yu and Frangakis, 2014*). Our approach differs by sampling random residues throughout the sequence and consequently provides higher precision in the estimation of template bias at high resolution. The observation that reconstructions with negligible template bias can be determined using particles identified with high-resolution template matching depends fundamentally on the noise model and threshold used to identify true positives and exclude false positives. We observe that the number of false positives does not perfectly match predictions based on a white Gaussian noise model, suggesting that the background is not perfectly Gaussian everywhere, for example due to local features with strong low-resolution contrast. When the noise model is uncertain or inaccurate, thresholding alone may not be sufficient to remove false positives. It is therefore important to validate features in reconstructions from targets found by template matching if they overlap with the template. In addition, overfitting could be assessed using the Omega metric described here, to quantify template bias in regions important for the study.

By further analogy to X-ray crystallography, the strategy we presented here could be extended by tiling through the template model, omitting overlapping features and combining the densities in each omitted region to form a continuous 3D map in which the density for each residue was omitted from the template, comparable in principle to a composite OMIT map (*Terwilliger et al., 2008*).

While currently computationally expensive and therefore not feasible in most cases, this strategy could be regarded as a 'gold standard', yielding reconstructions that are devoid of template bias while retaining the benefits of precise localization and identification of the targets. If only some map regions are validated, as was done in the examples presented here, it is likely that the rest of the 3D map is also reliable, based on the assumption that false positives were excluded from the reconstruction. However, this reasoning may not strictly hold when there is partial and variable mismatch between the targets and the template, for example due to conformational heterogeneity in the detected target population. In such a situation, template bias may not be uniform across the reconstruction, and template bias has to be assessed more rigorously.

## Materials and methods

### Yeast culture and FIB-milling

*S. cerevisiae* strains BY4741 (ATCC) colonies were grown to mid log phase in YPD, diluted to 10,000 cells/mL and treated with 10 µg/mL CHX (Sigma) for 10 min at 30°C with shaking as described in *Lucas and Grigorieff, 2023*. 3 µL were applied to a 2/1 or 2/2 Quantifoil 200 mesh $SiO_2$ Cu grid, allowed to rest for 15 s, back-side blotted for 8 s at 27°C, 95% humidity followed by plunge freezing in liquid ethane at –184°C using a Leica EM GP2 plunger. Frozen grids were stored in liquid nitrogen until FIB-milled. FIB-milling was performed as described in *Lucas and Grigorieff, 2023*.

### Cryo-EM data collection and image processing

Bgal micrograph movie data were downloaded from the EMPIAR database (EMPIAR-10644) and processed with the *cis*TEM image processing package (*Grant et al., 2018*) using Unblur (*Grant and Grigorieff, 2015*) to align and average the exposure-weighted movie frames, and CTFFIND4 (*Rohou and Grigorieff, 2015*) to determine image defocus values. Four of the 562 micrographs were discarded based on lack of clear CTF Thon rings or ice crystal contamination. The remaining 558 images were processed using *cis*TEM's template matching implementation (*Lucas et al., 2021*), yielding 59,259 targets with 2DTM SNRs above a threshold of 7.3.

Cryo-EM images of the yeast cytoplasm were previously published using imaging and processing pipelines as described in *Lucas and Grigorieff, 2023*, except that an additional seven images were included that were previously excluded because they contained organelle regions.

### Simulating 3D templates

The atomic coordinates from the indicated PDBs were used to generate a 3D volume using the *cis*TEM (*Grant et al., 2018*) program simulate (*Himes and Grigorieff, 2021*). For the Bgal template, we used a pixel size of 0.672 Å, which is slightly smaller than published for this dataset (0.68 Å). The smaller pixel size was obtained by fitting the 1.7 Å X-ray structure (PDB: 1DP0) into the published 2.2 Å cryo-EM map of PETG-bound Bgal, and adjusting the pixel size of the map to achieve optimal density overlap between model and map in UCSF Chimera (*Pettersen et al., 2004*). Details on template generation are summarized in *Table 1*.

**Table 1.** Preparation and simulation of the 3D templates used in this study.

| Template name | PDB | PDB modified? | Resolution of PDB map (Å) | PDB B-factor scaling | Additional B-factor applied (Å²) | Pixel size (Å) | Box size (pixels) |
|---|---|---|---|---|---|---|---|
| Bgal | 1DP0 | 10 Å sphere around Asp 102 deleted. HETATOMs excluded | 1.7 | 0 | 50 | 0.672 | 512 |
| LSU | 6Q8Y | Only atomic coordinates corresponding to the LSU included. HETATOMs excluded | 3.1 | 0 | 30 | 1.06 | 384 |
| LSU ($\Delta L7A$) | 6Q8Y | Only atomic coordinates corresponding to the LSU included. Atomic coordinates corresponding to L7A excluded. HETATOMs excluded | 3.1 | 0 | 30 | 1.06 | 384 |

## 2D template matching

2DTM was performed using the program *match_template* (**Lucas et al., 2021**) implemented in the *cis*TEM graphical user interface (**Grant et al., 2018**). For the Bgal searches, an in-plane angular step of 1.5° and an out-of-plane angular step of 2.5°, and D2 symmetry were used (no defocus search). This yielded a threshold of 7.30 calculated from a total number of ~$6.88 \times 10^{12}$ search locations, identifying targets with an average of one false positive per image.

For the LSU, an in-plane angular step of 1.5° and an out-of-plane angular step of 2.5°, and C1 symmetry and defocus search of ±1200 Å with a 200 Å step were used. This yielded a threshold of 7.85 calculated from a total number of ~$4.88 \times 10^{14}$ search locations, identifying targets with an average of one false positive per image.

## Generating 3D reconstructions

The *cis*TEM program prepare_stack_matchtemplate (**Lucas et al., 2021**) was used to generate particle stacks from the refined coordinates from the 2DTM searches followed by reconstruction using the *cis*TEM program reconstruct3d as described in the text. Local resolution estimation was performed using the local resolution tool in Phenix (**Liebschner et al., 2019**) using a box size of 7 Å (Bgal) or 12 Å (ribosome). To visualize regions of the ribosome reconstruction outside of the LSU template, we used the UCSF ChimeraX (**Pettersen et al., 2021**) volume tools to segment the map using a radius of 3 Å from the template atoms. UCSF Chimera (Bgal) (**Pettersen et al., 2004**) or ChimeraX (ribosome) (**Pettersen et al., 2021**) were used for visualization.

## Quantifying template bias

We wrote a program, measure_template_bias, which is part of the *cis*TEM software (**Grant et al., 2018**, source code available at https://github.com/timothygrant80/cisTEM, executables available at https://cistem.org/), to assess the degree of template bias present in a reconstruction, calculated from detected 2DTM targets. The program requires two templates on input, one template representing the full structure of the targets to be found (full template), and one containing omitted elements of the structure that serve as test regions to assess template bias (omit template). The program also requires the two reconstructions that were calculated form targets detected by these two templates (full reconstruction and omit reconstruction). The two templates and the two reconstructions have to be identically density-scaled, respectively. Using the two templates, measure_template_bias calculates a difference map that leaves only non-zero densities in areas omitted in the omit template. The difference map is then used as a mask to identify the test regions used to assess template bias. The densities in the test regions are summed for the two input reconstructions, yielding $\rho_{full}$ and $\rho_{omit}$, respectively. The average degree of template bias ($\Omega$) is then defined as the difference between $\rho_{full}$ and $\rho_{omit}$, relative to $\rho_{full}$ (**Equation 1**). $\Omega$ can assume values between 0 and 1 (100%), with 0 representing the least degree of template bias, and 1 representing the highest degree of template bias. If the degree of template bias has to be evaluated more locally, measure_template_bias also accepts a difference map, instead of the two templates, that will be used to identify the areas to be used for measuring template bias.

# Acknowledgements

The authors thank the members of the Grigorieff lab for helpful discussions. We are also grateful for the use of and support from the cryo-EM facility at UMass Chan Medical School. BAL and NG gratefully acknowledge funding from the Chan Zuckerberg Initiative, grant # 2021-234617 (5022).

# Additional information

### Competing interests

Bronwyn A Lucas, Benjamin A Himes: These authors are listed as inventors on a closely related patent application named "Methods and Systems for Imaging Interactions Between Particles and Fragments", filed on behalf of the University of Massachusetts. The patent relates to the use of the 2DTM method described in this manuscript, to image ligands and drugs bound to larger complexes that can

be detected by 2DTM. Nikolaus Grigorieff: This author is also listed as an inventor on a closely related patent application named "Methods and Systems for Imaging Interactions Between Particles and Fragments", filed on behalf of the University of Massachusetts. The patent relates to the use of the 2DTM method described in this manuscript, to image ligands and drugs bound to larger complexes that can be detected by 2DTM. Reviewing editor, eLife.

### Funding

| Funder | Grant reference number | Author |
|---|---|---|
| Howard Hughes Medical Institute | | Nikolaus Grigorieff |
| Chan Zuckerberg Initiative | 2021-234617 (5022) | Nikolaus Grigorieff |

The funders had no role in study design, data collection and interpretation, or the decision to submit the work for publication.

### Author contributions

Bronwyn A Lucas, Conceptualization, Data curation, Formal analysis, Funding acquisition, Validation, Investigation, Visualization, Methodology, Writing – original draft, Project administration, Writing – review and editing; Benjamin A Himes, Conceptualization, Formal analysis, Writing – review and editing; Nikolaus Grigorieff, Conceptualization, Software, Formal analysis, Supervision, Funding acquisition, Methodology, Writing – original draft, Project administration, Writing – review and editing

### Author ORCIDs

Bronwyn A Lucas http://orcid.org/0000-0001-9162-0421
Benjamin A Himes https://orcid.org/0000-0001-7777-0298
Nikolaus Grigorieff https://orcid.org/0000-0002-1506-909X

Reviewer #1 (Public Review): https://doi.org/10.7554/eLife.90486.3.sa1
Reviewer #2 (Public Review): https://doi.org/10.7554/eLife.90486.3.sa2
Reviewer #3 (Public Review): https://doi.org/10.7554/eLife.90486.3.sa3
Author Response https://doi.org/10.7554/eLife.90486.3.sa4

## Additional files

### Supplementary files

• MDAR checklist

### Data availability

All prior existing and new computer code used in this study is available at https://github.com/timothygrant80/cisTEM, (copy archived at *Timothygrant80, 2023*). Updated executables are available at https://cistem.org/.

The following previously published datasets were used:

| Author(s) | Year | Dataset title | Dataset URL | Database and Identifier |
|---|---|---|---|---|
| Saur M, Hartshorn MJ, Dong J, Reeks J, Bunkoczi G, Jhoti H, Williams PA | 2019 | Beta-galactosidase in complex with PETG | https://doi.org/10.6019/EMPIAR-10644 | EMPIAR, 10.6019/EMPIAR-10644 |
| Lucas BA, Grigorieff N | 2023 | Quantification of gallium cryo-FIB milling damage in biological lamella | https://doi.org/10.6019/EMPIAR-11544 | EMPIAR, 10.6019/EMPIAR-11544 |

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
