## [Editor Report · eLife assessment]

This is an **important** demonstration of how the false-positive rate of high-resolution 2D template matching to find particles of a given target structure in 2D cryo-EM images (2DTM) relates to overfitting the data towards the template. The authors present new methods to measure the amount of model bias that gets introduced in high-resolution features of such maps, with **compelling** evidence that high-resolution features that are not present in the template can still be reconstructed in 3D from images obtained by 2DTM.

---

## [Referee Report · Reviewer #1 (Public Review)]

This work continues a series of recent publications from the Grigorieff lab (https://doi.org/10.7554/eLife.25648, https://doi.org/10.7554/eLife.68946, https://doi.org/10.7554/eLife.79272, https://doi.org/10.1073/pnas.2301852120) showcasing the development of high-resolution 2D template matching (2DTM) for detection and reconstruction of macromolecules in cryo-electron microscopy (cryo-EM) images of crowded cellular environments. It is well known in the field of cryo-EM that searching noisy images with a template can result in retrieval of the template itself when averaging the candidate particles detected, an effect known as "Einstein-from-noise" (https://doi.org/10.1073/pnas.1314449110). Briefly, this occurs because it is statistically likely to find a match to an arbitrary motif over a large noisy dataset just by chance. The effect can be mitigated for example by limiting the resolution of the template, but this prevents the accurate detection of macromolecules in a crowded environment, as their "fingerprint" lies in the high-resolution range (https://doi.org/10.7554/eLife.25648). Here, the authors show through several experiments on in vitro and in situ data that features as small as drug compounds and water molecules can be reliably retrieved by 2DTM if they are searched by a template (the "bait") that contains expected neighboring features but not the targets themselves.

The ideas are generally clearly presented with appropriate references to related work, and claims are well supported by the data. In particular, the experiments for verifying the density of the ribosomal protein L7A as well as the systematic removal of residuals from the template model to assess bias are particularly clever.

The revised version of the manuscript addresses essentially all of the concerns raised previously by this reviewer, with the addition of figures and extended discussion of the key concepts.

---

## [Referee Report · Reviewer #2 (Public Review)]

This paper by Lucas et al follows on from earlier work by the same group. They use high-resolution 2D template matching (2DTM) to find particles of a given target structure in 2D cryo-EM images, either of in vitro single-particle samples or of more complicated samples, such as FIB-milled cells (which would otherwise perhaps be used for 3D electron tomography). One major concern for high-resolution template matching has been the amount of model bias that gets introduced into a reconstruction that is calculated straight from the orientations and positions identified by the projection matching algorithm. This paper assesses the amount of model bias that gets introduced in high-resolution features of such maps.

For a high-signal-to-noise in vitro single-particle cryo-EM data set, the authors show that their approach does not yield much model bias. This is probably not very surprising, as their method is basically a low false-positive particle picker, which works very well on such data. Still, I guess that is the whole point of it, and it is good to see that they can reconstruct density for a small-molecule compound that was not present in the original template.

For FIB-milled lamella of yeast cells with stalled ribosomes, the SNR is much lower and the dangers of model bias will be higher. This is also evidenced by the observation that further refinement of initial 2DTM identified orientations and positions worsens the map. This is obviously a more relevant SNR regime to assess their method. Still, they show convincing density for the GHX compound that was not present in the template, but was there in the reconstruction from the identified particles.

Quantification of the amount of model bias is then performed using omit maps, where every 20th residue in removed from the template and corresponding reconstructions are compared (for those residues) with the full-template reconstructions. As expected, model bias increases with lower thresholds for the picking. Some model bias (Omega=8%) remains even for very high thresholds. The authors state this may be due to overfitting of noise when template-matching true particles, instead of introducing false positive. Probably, that still represents some sort of problem. Especially because the authors then go on to show that their expectations of number of false positives do not always match the correct number of false positive, probably due to inaccuracies in the noise model for more complicated images, this may warrant further in-depth discussion in a revised manuscript.

Overall, I think this paper is well written and it has made me think differently (again) about the 2DTM technique and its usefulness in various applications, as outlined in the Discussion. Therefore, it will be a constructive contribution to the field.

After the first round of review, the authors addressed most points raised in a satisfying manner, which has led to a further (relatively minor) improvement of the manuscript.

---

## [Referee Report · Reviewer #3 (Public Review)]

The authors evaluate the effect of high-resolution 2D template matching on template bias in reconstructions and provide a quantitative metric for overfitting. It is an interesting manuscript that made me reevaluate and correct some mistakes in my understanding of overfitting and template bias, and I'm sure it will be of great use to others in the field.

The revised version of this manuscript addresses all of my concerns. The newly added Figure 4 supplement 1 provides a sobering outlook for the fraction of the proteome we can hope to identify in situ.

---

## [Author Response]

The following is the authors’ response to the original reviews.

The authors thank the reviewers for their thoughtful and constructive comments. We address each comment below and have uploaded a revised manuscript.

**Public Reviews**
1. One key point that could use further clarification is how to interpret densities in the reconstruction that do overlap with the template. If the omitted regions can be reliably reconstructed, and the density is smooth throughout, it implies the detected particles are not only (mostly) true positives but also their poses must be essentially correct. Therefore, why cannot the entire reconstruction be trusted, including portions overlapping with the template? In the "Future applications" section, the authors state that in order to obtain a reconstruction that is entirely devoid of template bias, it would be necessary to successively omit parts of the template structure through its entirety. I wonder if that is really necessary and if the presented approach of omitting template portions could be better framed as a "gold-standard" validation procedure.

Our assumption is indeed that the entire reconstruction can be trusted if the omitted features are faithfully reproduced in the reconstruction. We have added a sentence in the discussion to clarify this. However, we think that assessing template bias will still require the omit test (see also our reply below). Also, as discussed in the manuscript, there is likely a little bias left, even if it is not directly visible in the reconstruction. Therefore, if the goal is an entirely unbiased reconstruction, the only way will be to successively omit parts of the template structure throughout the template.

1. In other words, given the compelling evidence provided by the reconstructions in the omitted areas, I find it hard to imagine how the procedure would be "hallucinating" features in the rest of the structure, as the entire reconstruction depends on the same pose and defocus parameters. A possible experiment to test this hypothesis would be to go the opposite way, deliberately adding an unrealistic feature to the bait and checking whether it comes up in the reconstruction, while at the same time checking how it behaves in omitted parts.

Template bias might be generated in different ways. A common situation is the presence of noise, which causes biased deviations of the best template match from their “true” match that would just align the target signal to the template. Another type of bias may occur when there is a mismatch between the template and the detected target. The target may still be detected if there is sufficient structural overlap with the template. Since there might not be a clear “correct” alignment of a mismatching target to the template, the best alignment may again be biased, generating artificial density in the reconstruction. This second case may produce bias that is more pronounced in the mismatching regions. The different origins of bias will have to be investigated more thoroughly in another study. For the present study, however, we maintain that unless there is some assessment of bias in a given location, one cannot completely rule out bias based on the absence of it elsewhere in the reconstruction.

1. When assessing their approach to in situ data (the yeast ribosome), it is intriguing to see that the resolution downgraded from 3.1 to 8 Å when refinement of the particle poses against the current reconstruction was attempted. The authors do provide some possible explanations, such as the reduced signal of the reconstruction at high resolution and the crowded background, but it leaves one to wonder if this means that a 3.1 Å reconstruction could never be obtained from these data by conventional single-particle analysis procedures.

The refinement results with our in situ data do indeed appear to be limited to low resolution when using the conventional single-particle pipeline and software. It might be possible to improve refinement by introducing certain priors, filters and masking functions that are optimized for the increased background and spectral properties of in situ data. Also, we have not tested all available software, and some might perform better than others. It is worth noting that in a different study using our data, by Cheng et al (2023) and cited in our manuscript, the resolution of the refined reconstruction using different software was ~7 Å resolution, i.e., close to what we report here. Finally, refinement of the detected targets against a high-resolution template does work but since it involved the template, we regard this as part of the template matching process.

1. Furthermore, in the section "Quantifying template bias", the authors make the intriguing statement that there can still be some overfitting of noise even in true positives. I understand this overfitting would occur in the form of errors in the pose and defocus estimation, but a clarification would be helpful.

We have added a sentence in the Discussion to clarify where this bias may come from.

1. In the Discussion, the claim that "it is not necessary to use tomography to generate high-resolution reconstructions of macromolecular complexes in cells" is a misconception, at least in part. As demonstrated in works by the same group and others (https://doi.org/10.1016/j.xinn.2021.100166, https://doi.org/10.1038/s41467-023-36175-y, https://doi.org/10.1038/s41586-023-05831-0), 2D imaging of native cellular environments does offer a faster and better way to obtain high-resolution reconstructions compared to tomography. However, tomography provides the entire 3D context of the macromolecules, such as their localization to membranes and the cellular architecture, which can be readily visualized in a tomogram even at low resolution, so methods for structure determination from tilt series data such as subtomogram averaging remain of paramount importance. Most likely, a combination of 2D and 3D imaging approaches will be necessary to retrieve both the highest structural resolution and their cellular context to address biological questions.

We agree and have modified our statement accordingly.

1. The "Materials and Methods" section lacks a description of transmission electron microscopy data collection.

We are sorry for this oversight and have added these details.

1. Finally, the preprint version of this work posted on bioRxiv (https://doi.org/10.1101/2023.07.03.547552) contains the following competing interests statement, which is missing from the submitted version:"The authors are listed as inventors on a closely related patent application named "Methods and Systems for Imaging Interactions Between Particles and Fragments", filed on behalf of the University of Massachusetts."

This is correct. The statement was missing in the first version of the uploaded manuscript and was added after consultation with the eLife editorial office.

1. Quantification of the amount of model bias is then performed using omit maps, where every 20th residue is removed from the template and corresponding reconstructions are compared (for those residues) with the full-template reconstructions. As expected, model bias increases with lower thresholds for the picking. Some model bias (Omega=8%) remains even for very high thresholds. The authors state this may be due to overfitting of noise when template-matching true particles, instead of introducing false positives. Probably, that still represents some sort of problem. Especially because the authors then go on to show that their expectation of the number of false positives does not always match the correct number of false positives, probably due to inaccuracies in the noise model for more complicated images. This may warrant further in-depth discussion in a revised manuscript.

We have added further thoughts regarding the mismatch between expected and actual number of false positives in the Discussion section. A full understanding of the issue likely requires further study, which is currently underway.

1. The authors evaluate the effect of high-resolution 2D template matching on template bias in reconstructions, and provide a quantitative metric for overfitting. It is an interesting manuscript that made me reevaluate and correct some mistakes in my understanding of overfitting and template bias, and I'm sure it will be of great use to others in the field. However, its main point is to promote high-resolution 2D template matching (2DTM) as a more universal analysis method for in vitro and, more importantly, in situ data. While the experiments performed to that end are sound and well-executed in principle, I fail to make that specific conclusion from their results.

We do not see 2DTM as a more universal analysis method for in vitro and in situ data, but as simply as another method that can be used. We have added a sentence in the introduction to clarify this.

1. The authors correctly point out that overfitting is largely enabled by the presence of false-positives in the data set. They go on to perform their in situ experiments with ribosomes, which provide an extremely favorable amount of signal that is unrealistic for the vast majority of the proteome. This seems cherry-picked to keep the number of false-positives and false-negatives low. The relationship between overfitting/false-positive rate and the picking threshold will remain the same for smaller proteins (which is a very useful piece of knowledge from this study). However, the false-negative rate will increase a lot compared to ribosomes if the same high picking threshold is maintained. This will limit the applicability of 2DTM, especially for less-abundant proteins.

The reviewer is correct that the lower SNR of smaller targets poses a fundamental limit to 2DTM. We have stated this in previous studies and have added a sentence in the introduction of the current manuscript to clarify this.

1. I would like to see an ablation study: Take significantly smaller segments of the ribosome (for which the authors already have particle positions from full-template matching, which are reasonably close to the ground-truth), e.g. 50 kDa, 100 kDa, 200 kDa etc., and calculate the false-negative rate for the same picking threshold. If the resulting number of particles does plummet, it would be very helpful to discuss how that affects the utility of 2DTM for non-ribosomes in situ.

The suggested ablation study is a good idea and was reported by Rickgauer et al (2020), cited in our manuscript. We added our own analysis for this dataset in Figure 4-figure supplement 1 and show the proportion of LSUs detected as a function of template mass, indicating detection limit of ~300 kDa. We also added a note in the Results section to explain that the threshold we use to limit false positives means that there are also false negatives, with a rate that depends on their molecular mass.

1. Another point of concern is the dramatic resolution decrease to 8 A after multiple iterations of refinement against experimental reconstructions described in line 159. Was this a local search from the poses provided by 2DTM, or something more global? While this is not a manifestation of overfitting as the authors have conclusively shown, I think it adds an important point to the ongoing "But do we really need tomograms, or can we just 2D everything?" debate in the field, which is also central to the 2D part of 2DTM. Reaching 8 A with 12k ribosome particles would be considered a rather poor subtomogram averaging result these days. Being in the "we need tilt series to be less affected by non-Gaussian noise" camp myself, I wonder if this indicates 2D images are inherently worse for in situ samples. If they are, the same limitations would extend to template matching. In that case, shouldn't the authors advocate for 3DTM instead of 2DTM? It may not be needed for ribosomes, but could give smaller proteins the necessary edge.

We have extensively discussed the advantages and disadvantages of both tomography and 2DTM (Lucas et al, 2021) and think it is not useful to talk in terms of “better” and “worse”. Instead, each technique has its areas of application, and we maintain that a combination of the two may give the best results. The limitation of 8 Å does not apply to reconstructions aligned against high-resolution templates, as demonstrated in the present study. Regarding noise models, there is also need for these in 3DTM, as explained in recent publications: Maurer et al (2023), bioRxiv, doi.org/10.1101/2023.09.06.556487; Cruz-León et al (2023), bioRxiv, doi.org/10.1101/2023.09.05.556310; Chaillet et al (2023), Int. J. Mol. Sci. 24, 13375.

1. Right now, this study is also an invitation to practitioners who do not understand the picking threshold used here and cannot relate it to other template-matching programs to do a lot of questionable template matching and claim that the results are true because templates are "unoverfittable". I think such undesirable consequences should be discussed prominently.

We have added a discussion of this point in the Discussion section.

**Recommendations for the authors**
1. Lines 58-59: What does "nominally untilted" mean? Has the lamella pre-tilt (milling angle) been taken into account or not? If yes, how?

The lamella milling angle was not taken into account, so there is a tilt built into the sample of about 8° that was not compensated for by a counter-tilt of the microscope goniometer. We have added a note to explain this in the text of the manuscript.

1. Lines 113-114: A brief explanation of the threshold calculation method from Rickgauer et al, 2017 to achieve an expected false positive rate of one per micrograph would be helpful here.

We describe the equation for estimating the false discovery rate later in the manuscript. We have added a note in the text to point the reader to the relevant section of the manuscript.

1. For consistency, it would be interesting to include a plot of the SNR peaks found by 2DTM in the in situ dataset, that could be directly compared to Figure 1 - figure supplement 1B.

We have added this to Figure 2 - figure supplement 1A-C, to directly compare to Figure 1 – figure supplement 1A-C.

1. Showing model-map FSC curves between the density retrieved from the omitted areas and their respective models would provide further evidence not only that they are correct but to what extent.

An FSC calculation would be challenging for small regions, such as side chains and drugs, due to masking artifacts. Moreover, the model was built into an in vitro determined map and was not fit into the in vivo map calculated here. Therefore, deviations between the map and model may reflect differences between the two conditions and may not reflect the agreement of the map to the in vivo structure.

1. Lines 128-130: The figure references are wrong. Here, Figure 1B should probably be Figure 1A (or 1B), and Figure 1C clearly refers to Supplementary Figure 1F (FSC curve).

We have corrected the incorrect figure references.

1. Line 125: Wrong figure reference, Figure 1A here refers to Supplementary Figure 1B (cross-correlation peaks).

We have corrected the incorrect figure references.

1. I haven't been able to find mention of code availability in the manuscript. Given that it is a major outcome of the study, I think it should be provided.

The code is available from the cisTEM repository, github.com/timothygrant80/cisTEM, and an executable version of the program measure_template_bias has been posted for download on the cisTEM webpage, cistem.org. We have added a note in the Methods section to point the readers to these resources.

1. Line 50: "An additional complication of subtomogram averaging for in situ imaging is the selection of valid targets" - This is not specific to subtomogram averaging, but to in situ samples.

We agree and have updated the text to reflect this.

1. Line 77: "if this is true for high-resolution features, which are more susceptible to noise overfitting" - This is not intuitive to me. High-resolution features require more information to be overfitted with a constant set of model parameters, thus making their overfitting harder.

The reviewer is correct that there is more information at high resolution, partially compensating for the low SNR. However, the overall refinement behavior is still dominated by overfitting at high resolution, as we have demonstrated in an earlier publication in Stewart & Grigorieff (2004), Ultramicroscopy 102, 67–84.

1. Line 316: "Baited reconstruction is substantially faster and a more streamlined" - To back this and other similar statements, it would be helpful if the authors provided some time measurements for the execution of their potentially very computationally expensive search.

The current implementation of 2DTM requires 45 GPU hours per template per K3 image to search 13 defocus planes. However, for a comparison, the manual work for annotation, as well as additional processing to align and classify sub-tomograms to generate high resolution averages should also be considered in this comparison. These are highly project-dependent and can exceed the time required for 3DTM manifold. We have clarified this in our Discussion section.

1. Line 319: "We expect focused classification to identify sub-populations to further improve the resolution" - How would this work if refining the 2D data without a high-resolution template resulted in significantly worse resolution even for a ribosome? Or is this meant to be done with prior knowledge of every state?

Classification can be done using existing single particle software. To avoid alignment errors, as described above, particle alignment angles and shifts are fixed during classification. This leaves only the particle occupancy per class to be refined, which appears to lead to good classification. We have added a brief note to explain this strategy. However, since this is not shown in this manuscript, we have not added a more extensive discussion of particle classification.

1. Line 354: "without requiring manual intervention or expert knowledge" - Previous expert knowledge was arguably provided in the form of a high-resolution structure.

We agree with the reviewer and have clarified our statement.